# Incongruent virtual reality cycling exercise demonstrates a role of perceived effort in cardiovascular control

Richard M. Bruce[1] , Gerrard F. Rafferty[1], Sarah L. Finnegan[2], Martin Sergeant[3], Kyle T.S. Pattinson[2] and Oliver R. Runswick[4]

[1]*Centre for Human and Applied Physiological Sciences, School of Basic and Medical Biosciences, Faculty of Life Science and Medicine, King's College London, London, United Kingdom*
[2]*Nuffield Department of Clinical Neurosciences, University of Oxford, Oxford, United Kingdom*
[3]*MRC Weatherall Institute of Molecular Medicine, University of Oxford, Oxford, United Kingdom*
[4]*Department of Psychology, Institute of Psychiatry, Psychology and Neuroscience, King's College, London, United Kingdom*

Handling Editors: Vaughan Macefield and Yoshihiro Kubo

The peer review history is available in the Supporting Information section of this article (https://doi.org/10.1113/JP287421#support-information-section).

The Journal of Physiology

**Abstract figure legend** Participants cycled through virtual reality environments comprising a flat road and differently graded inclines (3%, 6%, 9%). During one visit the pedalling resistance was identical despite the different virtual hill gradients. Steeper virtual hills transiently augmented perceptions of exertion and cardiovascular responses, although the opposite was not found with less-steep hills.

**Richard M. Bruce** is a senior lecturer in cardiorespiratory physiology at King's College London. He completed his PhD at the University of Birmingham and much of his postdoctoral work at the University of Oxford. His research interests are in human cardiorespiratory control in health and disease and developing new technologies to assess cardiovascular and respiratory functions.

**Abstract** In this study we have used a highly immersive virtual reality (VR) cycling environment where incongruence between virtual hill gradient (created by visual gradient and bike tilt angle) and actual workload (pedalling resistance) can experimentally manipulate perception of exercise effort. This therefore may provide a method to examine the role of effort perception in cardiorespiratory control during exercise. Twelve healthy untrained participants (7 men, age $26 \pm 5$ years) were studied during five visits. On visit 1 participants underwent cardiopulmonary exercise testing, and during subsequent visits (2–4) participants performed repeated hill climbs at different gradients (of 3%, 6% and 9% in counterbalanced order) with the actual workload 'congruent' with virtual hill gradient. On visit 5 participants completed three incongruent trials with virtual hill gradients of 3%, 6% and 9% but a fixed workload equal to that for the 6% climb (iVR3%, iVR6% and iVR9% trials). Despite no difference in power output, there was a significantly elevated rating of perceived exertion (RPE) and mean arterial blood pressure in iVR9% compared to iVR3% and iVR6%, although this effect decayed over time. There was no effect on any respiratory variable, and no significant reduction in RPE or cardiovascular responses was observed during the iVR3% trial. These data suggest that perception of effort and cardiovascular responses to exercise can be manipulated experimentally via virtual hill gradient (using visual and/or vestibular cues) in a VR environment. This work supports those previously showing the existence of a control mechanism which integrates perception of effort and the cardiovascular response to exercise in humans.

(Received 31 July 2024; accepted after revision 9 December 2024; first published online 4 January 2025)

**Corresponding author** R. M. Bruce: Centre for Human and Applied Physiological Sciences, School of Basic and Medical Biosciences, Faculty of Life Science and Medicine, King's College London, Guy's Campus, London SE1 1UL, UK. Email: richard.bruce@kcl.ac.uk

**Key points**

- We aimed to assess whether using a highly immersive virtual reality (VR) cycling environment to create incongruence between perceived effort (virtual hill gradient) and actual effort (pedal resistance) can manipulate cardiorespiratory responses to exercise.
- At an equivalent power output cycling up a steeper virtual hill produced greater ratings of perceived exertion (RPEs) and blood pressure responses compared to a virtual hill congruent to power output.
- This work suggests the existence of a control mechanism which integrates perception of exercise effort and the cardiovascular response to exercise, which can be experimentally manipulated by VR.

## Introduction

At the onset of dynamic exercise, oxygen delivery rapidly increases to active muscular tissue due to the integrated responses of the cardiovascular and respiratory systems. The control mechanisms driving these initial cardio-respiratory responses are incompletely understood, but it is widely thought that both neural afferent feedback from exercising skeletal muscle and feedforward signals from higher brain centres contribute (Bruce et al., 2019; Fisher et al., 2015; Forster et al., 2012). The feedforward mechanism, often termed 'central command', is classically defined as the parallel activation of motor and cardio-respiratory centres in the brain resulting in the gross matching of muscle oxygen delivery to exercise workload (Goodwin et al., 1972).

Despite much pioneering work investigating central command control of the cardiorespiratory system (e.g. Eldridge et al., 1981; Green et al., 2007; Thornton et al., 2001), its neuro-circuitry remains poorly understood, and it is likely that other factors, such as perception of effort, need to be built into a more complex model (Williamson, 2010). Perceived effort during exercise (also commonly called 'perceived exertion' or 'effort sense') is regarded as the 'conscious sensation of how hard, heavy, and strenuous a physical task is' (Marcora, 2010b). The underlying mechanism is, however, still debated. Although the involvement of top-down signalling (corollary discharge from motor areas) is well established, the role and integration of bottom-up signalling (sensory feedback) are still argued (Abbiss et al., 2015; Amann & Secher, 2010;

Marcora, 2010a, 2010b; Monjo & Allen, 2023; Pageaux, 2016). It is also likely that the process is regulated by higher-order cognitive factors such as knowledge of task difficulty and memory of previous experience (Abbiss et al., 2015; Marcora, 2010b).

It is therefore possible that higher-order factors form part of a 'central command' cardiorespiratory control mechanism in exercise. Indeed, innovative experimental designs have shown that imagined exercise (Williamson et al., 2002), or watching video footage of first-person 'point-of-view' exercise (Brown et al., 2013), can generate cardiorespiratory responses approximating those produced during active exercise. Furthermore the cardiorespiratory responses to exercise can be altered by hypnotic suggestion of performing exercise at higher or lower workloads *versus* reality (Morgan et al., 1973; Williamson et al., 2001). The advantage of this design is that exercise workload (and hence muscle afferent feedback) can be fixed allowing the 'central command' mechanism to be studied independently. Despite the significance of this work, there are potential limitations to using hypnosis experimentally. Firstly not all individuals are 'hypnotisable', so it is possible that selection/sampling bias exists in this type of experimental approach. Furthermore it is unclear what neurophysiological mechanisms are involved in the perception of effort under hypnosis and how these relate to those during real exercise experiences. It is possible that other experimental models, such as the manipulation of virtual reality (VR) environments, have a more universal effect and greater real-world application (Finnegan et al., 2023; Runswick et al., 2023).

Using VR individuals can be placed in an immersive environment which can be controlled and manipulated independently from real-world experiences. The application of this to exercise paradigms has been previously demonstrated with a highly immersive VR cycling environment (Finnegan et al., 2023). By adjusting the gradient of virtual hills, in combination with a cycle ergometer which can be tilted to simulate hill gradients and a fan to simulate the movement through the air across a range of speeds, we aim to manipulate participants' perception of effort independently of actual workload (pedalling resistance). In this way multiple sensory inputs can be manipulated (visual, vestibular, tactile) as they might occur in the real world. We have previously shown that compared to non-VR laboratory environments, VR increases chosen cycling workload with no change in effort perception (Runswick et al., 2023). In addition using VR to create incongruence between perceived hill gradient and actual pedalling resistance alters perceptions of effort. Simulating increased or decreased inclines resulted in elevations and reductions in ratings of perceived exertion/breathlessness, respectively (Finnegan et al., 2023; Runswick et al., 2023). However as cycling workload was not fixed between interventions (VR and non-VR environments) in our previous studies, it is unclear (i) if VR modulates changes in exercise perceived effort independently of actual workload and (ii) how this impacts the physiological response to exercise. This is the focus of the current study.

This investigation aims to examine whether the perception of effort alters cardiorespiratory responses to exercise independently of actual workload. This will be achieved by simulating different uphill gradients incongruent with a fixed cycling workload and assessing the rating of perceived exertion (RPE) and the cardio-respiratory responses during the exercise. We hypothesise that at a fixed workload a greater simulated incline will generate augmented RPE and cardiorespiratory responses, whereas a lowered incline will have the opposite effect.

## Methods

### Ethical approval and participants

Twelve healthy participants (7 men, age $26 \pm 5$ years, height $1.76 \pm 0.07$ m, body mass $73.3 \pm 11.7$ kg, maximal oxygen consumption $39.9 \pm 7.7$ ml/kg/min) participated in the study after providing written informed consent. Participants were required to be $\geq 18$ years with no history of metabolic, respiratory or cardio-vascular disease. All participants had experience of cycling, but all were untrained and cycled an average of 0–1 h per week. Participants were unaware of the nature of the experimental manipulation of the VR environment (see later). All the experimental procedures conformed to the latest revision of the *Declaration of Helsinki*, except for registration in a database, and were approved by a local university research ethics committee (LRS/DP-21/22-26409).

### Study design

Participants visited the laboratory on five occasions, with >72 h between each visit. During visit 1 participants were familiarised to all procedures, and an exclusion criterion was the development of any motion sickness during VR familiarisation. No motion sickness was reported. During visit 1 participants also underwent a maximal incremental cycling exercise task to assess ventilatory threshold 1 (VT1) and maximal oxygen consumption ($VO_2$max). During visits 2–4 participants performed six submaximal cycling exercise trials within a VR environment, first described by Finnegan et al (2023) and in the provisional patent (Pattinson & Finnegan, 2022), that is, cycling through a VR course with flat and uphill sections. Full details are described in the protocol later, but in brief these involved cycling up different hill gradients (3%,

6% and 9%), with pedalling resistances 'congruent' to the respective VR hill gradient (cVR3%, cVR6% and cVR9%). Visit 5 was the experimental aspect of the study where the responses to 'incongruent' VR (iVR) cycling were assessed. This involved three cycling trials, all with the same pedalling resistance but employing three different VR hill gradients (iVR3%, iVR6% and iVR9%). Participants were asked to refrain from consuming food and caffeine within 4 h and performing strenuous physical activity or consuming alcohol within 24 h of each visit. The study followed a repeated measures design with all participants performing all trials, with the experimental hill gradients in visit 5 (3%, 6% and 9%) performed in a counterbalanced order.

## Protocol

Visit 1: VT1 and VO2max and maximum power output (Wmax) were determined using an incremental ramp cycling exercise task, with the ramp (Watts/min) determined from standard prediction equations (Wasserman et al., 2005) and predicted Wmax achieved in 8–12 min. $VO_2$max was calculated as the highest-achieved mean oxygen consumption across a 20 s period, and VT1 was determined using the V-slope method with support from the ventilatory equivalent for oxygen and the respiratory exchange ratio (as described by Beaver et al., 1986).

Visits 2–4: six submaximal cycling exercise trials were performed within a VR environment. Each trial involved participants cycling along a straight road consisting of six stages: $3 \times 500$ m flat stages (0% incline) interspersed with $3 \times 500$ m hill stages at different gradients – 3%,

6% and 9%. The pedalling resistances were congruent with the virtual hill (visual gradient and bike tilt), and the fan speed adjusted automatically to match cycling speed. The order of the hill gradients was counterbalanced across the sample. The order remained fixed through all subsequent testing for each participant. Participants were asked to maintain 60RPM, and instantaneous visual feedback on RPM was provided to participants by a virtual display of RPM in the VR environment. Across visits 2–4 participants completed 54 separate conditioning hill climbs, designed to allow participants to associate each hill gradient with a certain pedalling resistance and physiological response.

Visit 5 formed the experimental portion of the study, and physiological variables were recorded. Participants first performed three congruent trials (**cVR3%**, **cVR6%** and **cVR9%**) in a counterbalanced order (Fig. 1). Each trial required participants to sit at rest on the bike for 1 min and then cycle along a straight road with two stages: a flat stage (0% gradient) and a hill stage (at 3%, 6% or 9%; see Fig. 1). The distances covered through the flat and hill stages were adjusted to ensure that participants cycled on them for at least 4 min each. The pedalling resistances were congruent with the virtual hill (visual gradient and bike tilt), and the fan speed adjusted automatically to match cycling speed. Participants were asked to maintain 60RPM throughout, and a 15 min rest was provided between trials. The exact pedalling resistance, and hence the power output, was calibrated to the **cVR6%** trial. The pedalling resistance during the 6% trials was adjusted for each participant to generate a power output approximately equal to that achieved at 90% VT1 (when cycling at 60RPM). Participants then performed three incongruent

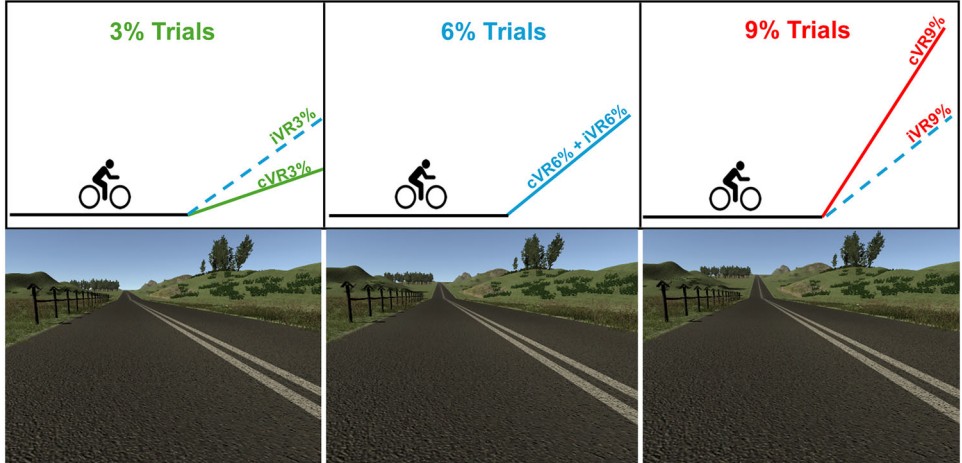

**Figure 1. The 3 congruent and 3 incongruent trials.**
The solid lines represent the VR (virtual reality) hill gradient (3%, 6% and 9% gradients), and during the 3 congruent trials (cVR3%, cVR6% and cVR9%) pedalling resistance matched the hill gradient. During the 3 incongruent trials (iVR3%, iVR6% and iVR9%), the same VR hill gradients were completed, but the pedalling resistances were identical between the trials (set at the resistance congruent with a 6% hill). This is represented by the broken lines. Note that the angle of the lines is illustrative and do not exactly match the actual gradient.

trials (**iVR3%**, **iVR6%** and **iVR9%**) in the same counter-balanced order. Participants were not informed of the incongruence. The protocol was identical to that used in the **cVR3%**, **cVR6%** and **cVR9%** trials, respectively, except the pedalling resistance was fixed at that for the **cVR6%** gradient trial (∼90% VT1). This is represented by dotted lines in Fig. 1.

To assess whether the different bike tilt angle (independent of the visual gradient) alters the physiological responses to exercise, participants completed a final non-VR cycling trial (**nVR**) where cycling at 3 × 3 min periods at ∼90%VT was performed in the laboratory environment, with the bike tilted to 3%, 6% and 9% (**nVR3%, nVR6% and nVR9%**) in a counter-balanced order.

### Equipment and measurements

VT1, $VO_2$max and Wmax were assessed using a cycle ergometer (Lode Excalibur Sport, Groeningen, the Netherlands) and an automated metabolic cart system (Metalyzer 3B, Cortex, Leipzig, Germany), which was calibrated prior to each study per manufacturer's instructions.

VR cycling exercise was undertaken using a different automated cycle ergometer (Wahoo-Kickr Climb Integrated Cycling system, Wahoo, Atlanta, GA, USA) and a custom-made VR environment written in Unity-3D. The simulation was rendered in a VR headset (Vive XR Elite, HTC, Taiwan) connected to a VR-ready PC (3XS High-End Gaming PC with NVIDIA GeForce RTX 3080 and AMD Ryzen 7 5800X, Santa Clara, CA, USA). The incline (up/down tilt) of the Wahoo cycle ergometer adjusted to match the gradient of the virtual hill/flat. The use of Wahoo cycle ergometers has been validated previously (Gin et al., 2018). In addition a fan (Wahoo-Kickr headwind, Wahoo, Atlanta, GA, USA) was placed directly in front of the bike and automatically adjusted its air flow to match the estimated cycling speed. More details of the setup can be found elsewhere (Finnegan et al., 2023; Runswick et al., 2023).

During the cVR, iVR and nVR trials of visit 5, respiratory data (Metalyzer 3B, Cortex, Leipzig, Germany) and heart rate (HR) data (Polar H10. Polar Electro, Kempele, Finland) were recorded continuously. During cVR and iVR, brachial artery systolic (SBP) and diastolic (DBP) blood pressure was recorded intermittently using an automated sphygmomanometer (Tango M2 stress test BP monitor, Suntech Medical, Morrisville, NC, USA) from the non-dominant arm at rest, during the final minute of the flat stage and at 0.5 and 3.5 min of the hill stage. During familiarisation participants were instructed to grip the handlebars loosely with their non-dominant hand when the blood pressure was recorded. Blood pressure

was measured only on three occasions during exercise to minimise the impact of the measurement procedure on VR immersion. Participants reported their RPE (Borg CR10 scale; Borg, 1982) verbally during the final minute of the flat stage, and at 0.5, 1.5, 2.5 and 3.5 min of the hill stage. During nVR blood pressure and RPE were recorded during the final minute of each hill gradient stage.

### Data analysis

All data were analysed and presented using GraphPad Prism (GraphPad Software Inc, version 10.2.1, Boston, MA, USA) and a standard statistical package (SPSS, version 22, IBM, Chicago, IL, USA). Heart rate, minute ventilation, oxygen consumption ($VO_2$) and power output were calculated in 30 s epochs throughout the trial stages. Mean arterial blood pressure (MABP) was calculated as DBP + (SBP – DBP)/3. All data are presented as mean and standard deviation unless otherwise stated. For **cVR** and **iVR** differences between trials and between trial stages were analysed using a two-way repeated-measures ANOVA, and for **nVR** differences between trials were analysed using a one-way repeated-measures ANOVA. Where appropriate *post hoc* analysis was performed using a Bonferroni correction. Before ANOVA if Mauchly's test of sphericity was violated, degrees of freedom were adjusted in accordance with the Greenhouse–Geisser test. Correlation between continuous variables was assessed using Spearman's rank correlation coefficient ($\rho$). Statistical significance was taken as $p < 0.05$.

### Results

#### cVR and iVR trials: Power output and respiratory variables

Figure 2 shows the changes in power output (W), $VO_2$ (l/min) and ventilation (l/min) throughout the different stages of the congruent (**cVR3%, cVR6%** and **cVR9%**) and incongruent (**iVR3%, iVR6%** and **iVR9%**) trials. For the congruent trials there was a significant interaction effect between trial and stage for power output [$F_{(3.5, 36.7)} = 357.3$, $p < 0.001$] $VO_2$ [$F_{(4.5, 49.6)} = 101.6$, $p < 0.001$] and ventilation [$F_{(4.1, 45.3)} = 71.7$, $p < 0.001$] where values from all three trials were significantly different from each other during all hill stages (0.5–4 min). For incongruent trials there was no significant interaction or main effect for trial, but there was a significant main effect for stage with power output [$F_{(1.1, 12.9)} = 597.1$, $p < 0.001$], $VO_2$ [$F_{(2.1, 22.8)} = 410.1$, $p < 0.001$] and ventilation [$F_{(2.3, 25.7)} = 265.2$, $p < 0.001$].

For the congruent trials there was a significant interaction effect between trial and stage for breathing frequency [$F_{(6.03, 66.3)} = 5.04$, $p < 0.001$] and tidal

volume [F(3.86, 42.4) = 39.9, $p < 0.001$]. Breathing frequency was significantly higher in **cVR9%** than **cVR3%** and in **cVR9%** than **cVR6%** during all hill stages (0.5–4 min), but there were no differences between **cVR3%** and **cVR6%** trials. Tidal volumes from all three trials were significantly different from each other during all hill stages (0.5–4 min). For incongruent trials there was no significant interaction or main effect for trial, but there was a significant main effect for stage with breathing frequency [F(2.5, 28) = 34.7, $p < 0.001$] and tidal volume [F(1.7, 18.77) = 164.5, $p < 0.001$].

## cVR and iVR trials: Perception and cardiovascular variables

Figure 3 shows the changes in RPE, HR (bpm) and MABP (mmHg) throughout the different stages of the congruent (**cVR3%, cVR6%** and **cVR9%**) and incongruent (**iVR3%, iVR6%** and **iVR9%**) trials. For the congruent trials there was a significant interaction effect between trial and stage for RPE [F(4.3, 43.4) = 43.6, $p < 0.001$], HR [F(4.3, 47.6) = 120, $p < 0.001$] and MABP [F(2.4, 26.5) = 14.3, $p < 0.001$] where values from all three trials were significantly different from each other during

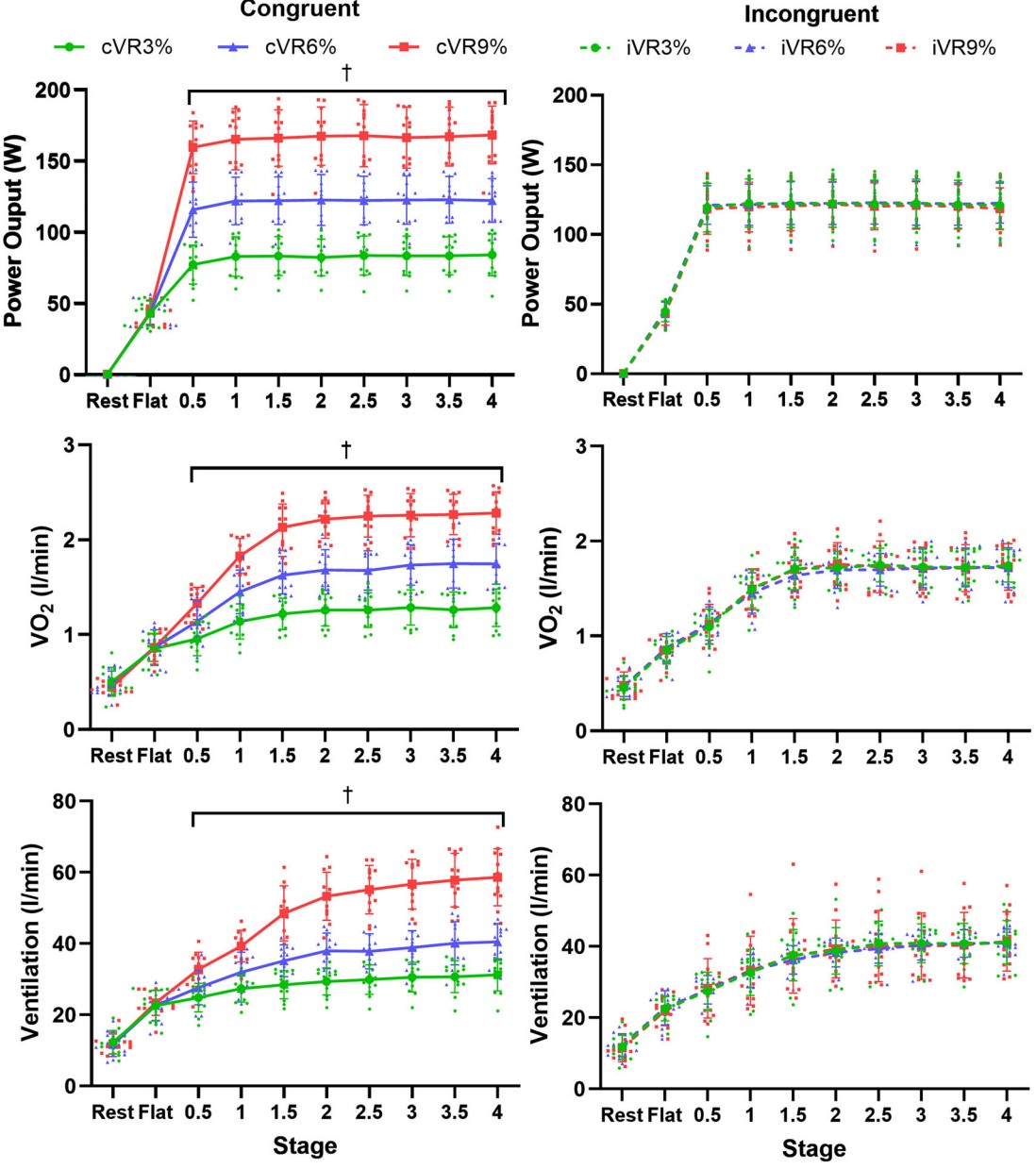

**Figure 2. The change in mean (±SD) power output (*A* and *B*), oxygen consumption (VO$_2$, *C* and *D*) and ventilation (*E* and *F*) throughout the congruent (solid lines) and incongruent trials (dotted lines).**
†, significant difference between **cVR3%** and **cVR6%**, **cVR3%** and **cVR9%** and **cVR6%** and **cVR9%**.

all hill stages. There was a significant interaction effect between trial and stage for SBP [$F_{(6, 66)} = 25.3$, $p < 0.001$] where values from all three trials were significantly different from each other during all hill stages (at 0.5 min, **cVR3% =** $140.8 \pm 12.5$, **cVR6% =** $145.8 \pm 10.4$ and **cVR9% =** $158.5 \pm 10.9$ mmHg; at 3.5 min, **cVR3% =** $140.5 \pm 11.5$, **cVR6% =** $150.3 \pm 11.1$ and **cVR9% =** $164.2 \pm 10.9$ mmHg). There was no significant difference in SBP at any other time point. There was also

a significant interaction effect between trial and stage for DBP [$F_{(2.69, 29.6)} = 4.09$, $p = 0.018$], where values on hill stages were significantly different between **cVR9%** and **cVR6%** (at 0.5 min, $92.8 \pm 10.4$ *vs.* $86.1 \pm 9.7$ mmHg; at 3.5 min, $93.7 \pm 10.9$ *vs.* $89.6 \pm 9.1$ mmHg) and between **cVR9%** and **cVR3%** (at 0.5 min, $92.8 \pm 10.4$ *vs.* $82.4 \pm 10$ mmHg; at 3.5 min $93.7 \pm 10.9$ *vs.* $84.2 \pm 10.6$ mmHg). There were no significant differences in DBP between **iVR6%** and **iVR3%** or at any other time point between any

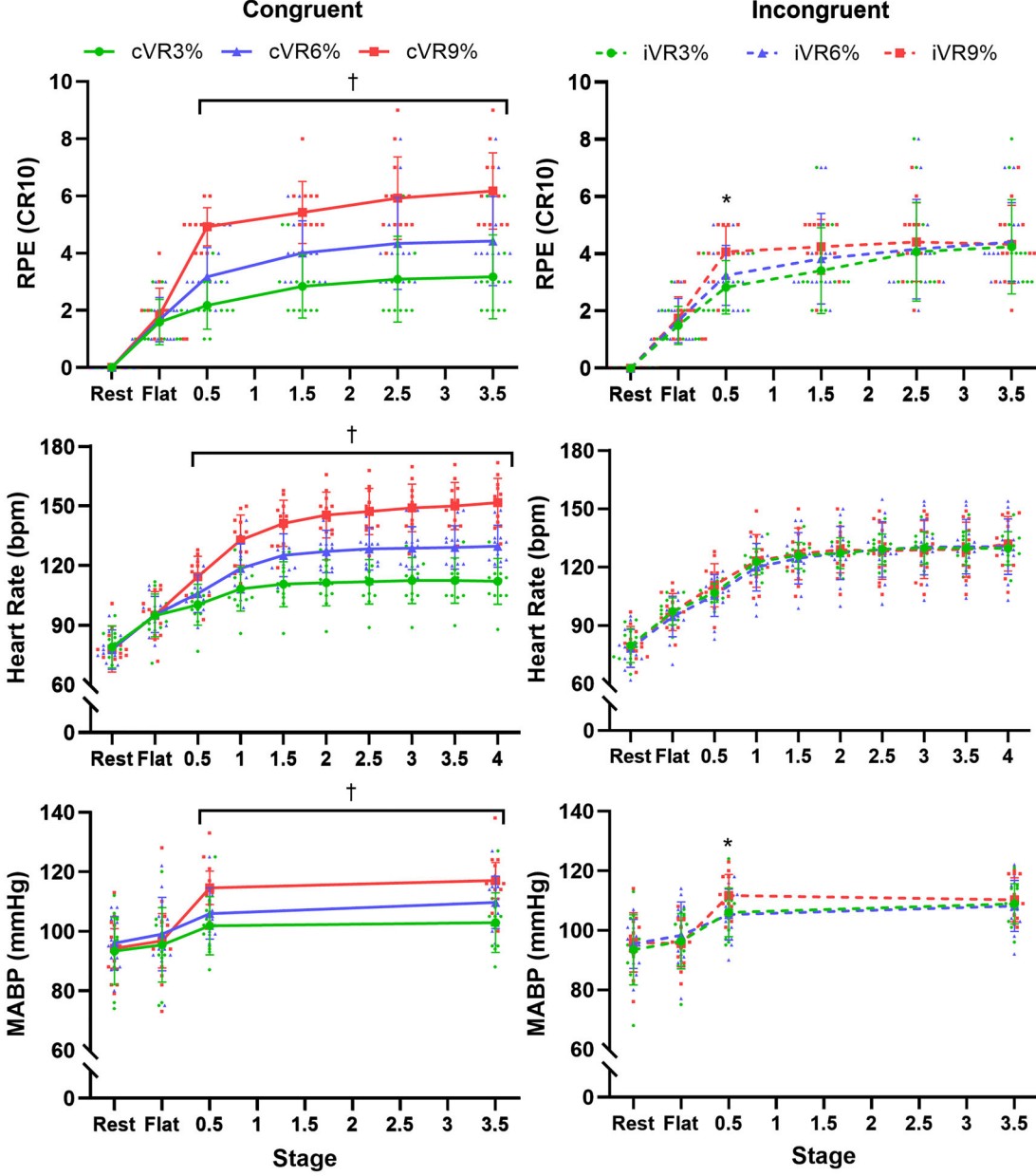

**Figure 3. The change in mean (±SD) rating of perceived exertion (RPE, *A* and *B*), heart rate (*C* and *D*) and mean arterial blood pressure (MABP, *E* and *F*) throughout the congruent (solid lines) and incongruent trials (dotted lines).**
†, significant difference between **cVR3%** and **cVR6%**, **cVR3%** and **cVR9%** and **cVR6%** and **cVR9%** ($p < 0.05$).
*, significant difference between **iVR3%** and **iVR9%** and **iVR6%** and **iVR9%** ($p < 0.05$).

trials. For incongruent trials there was a significant interaction between trial and stage for RPE [$F_{(2.7, 29.6)} = 3.5$, $p = 0.017$] where RPE was significantly higher at 0.5 min in **iVR9%** than **iVR6%** ($4.08 \pm 1$ *vs.* $3.25 \pm 1, p = 0.01$) and in **iVR9%** than **iVR3%** ($4.08 \pm 1$ *vs.* $2.83 \pm 0.9, p < 0.001$), but there was no difference between **iVR6%** and **iVR3%** at 0.5 min, or any other differences at any time point. In incongruent trials although there was no significant interaction between trial and stage for HR [$F_{(3.8, 42.2)} = 1.38$, $p = 0.26$], significant interaction between trial and stage was observed for MABP [$F_{(6, 66)} = 2.79$, $p < 0.013$], with MABP significantly higher at 0.5 min in **iVR9%** than **iVR6%** ($111.6 \pm 9.1$ *vs.* $105.4 \pm 8.6$ mmHg, $p < 0.032$) and in **iVR9%** than **iVR3%** ($111.6 \pm 9$ *vs.* $106 \pm 8.3$ mmHg, $p < 0.018$). There were no significant differences in MABP between **iVR6%** and **iVR3%** at 0.5 min, or any other time point. There was no significant interaction between trial and stage for DBP [$F_{(6, 66)} = 0.54, p = 0.77$], but there was a significant interaction between trial and stage for SBP [$_{(6, 66)} = 2.34, p = 0.041$], with SBP significantly higher at 0.5 min in **iVR9%** than **iVR6%** ($152.3 \pm 13$ *vs.* $142.8 \pm 10.7$ mmHg, $p = 0.014$) and in **iVR9%** than **iVR3%** ($152.3 \pm 13$ *vs.* $144.9 \pm 10.7$ mmHg, $p = 0.043$). There were no significant differences in SBP between **iVR6%** and **iVR3%** at 0.5 min or any other time point.

Figure 4 shows the linear relationships between the magnitude of the experimental effect on MABP *versus* RPE and HR *versus* RPE. Specifically, the analysis has been performed by first calculating the difference (Diff) in MABP, HR and RPE recorded between trials at 0.5 min (iVR9% – iVR6% and iVR9% – iVR3%). Then the correlation between $MABP_{Diff}$ *versus* $RPE_{Diff}$ (A) and $HR_{Diff}$ *versus* $RPE_{Diff}$ (B) was assessed. These data aim to provide insight into whether the significant effects described in Fig. 3 (for RPE and MABP) are associated with each other or just independent effects. There

was a significant moderate positive correlation between $MABP_{Diff}$ and $RPE_{Diff}$ ($\rho = +0.39$, $p = 0.031$) and also between $HR_{Diff}$ and $RPE_{Diff}$ ($\rho = +0.48$, $p = 0.012$).

### nVR trials: All variables

Figure 5 shows the power output (W), $VO_2$ (l/min), ventilation (l/min), RPE, HR (bpm) and MABP (mmHg) during cycling at three hill gradients (**nVR3%, nVR6%** and **nVR9%**), and in the laboratory 'non-VR' environment. For all variables, there were no significant differences between any of the trials.

### Discussion

This aim of this investigation was to examine whether manipulating the perception of exercise effort, via the VR simulation of different hill gradients, could alter cardiorespiratory responses independently of actual effort – a fixed cycling workload. We have demonstrated that cycling up a steeper virtual hill resulted in a transient elevation in RPE and blood pressure despite actual workload remaining unchanged. This work therefore presents a novel approach to examine 'central command' cardiorespiratory control mechanism in human exercise. Although some findings did not agree with our original hypothesis, the results suggest perception of effort (RPE) and cardiovascular responses to exercise can be manipulated experimentally via virtual hill gradient (using visual and/or vestibular cues) in a VR environment. The current data support previous studies that demonstrate the existence of a control mechanism which integrates perception of effort and cardiovascular response to exercise in humans.

These findings further contribute to the body of literature which demonstrates the role of central

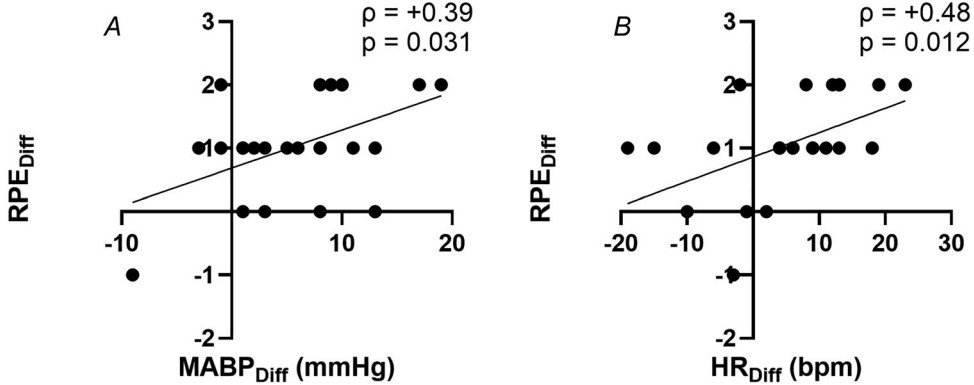

**Figure 4. The correlation between the magnitude of the experimental effect on MABP (mean arterial blood pressure) and RPE (rating of perceived exertion) and between HR and RPE.**
Specifically the analysis was performed by calculating the difference (Diff) in MABP, HR and RPE recorded between trials at 0.5 min (iVR9% – iVR6% and iVR9% – iVR3%), and then the correlation between $MABP_{Diff}$ and $RPE_{Diff}$ (A) and between $HR_{Diff}$ and $RPE_{Diff}$ (B) was assessed.

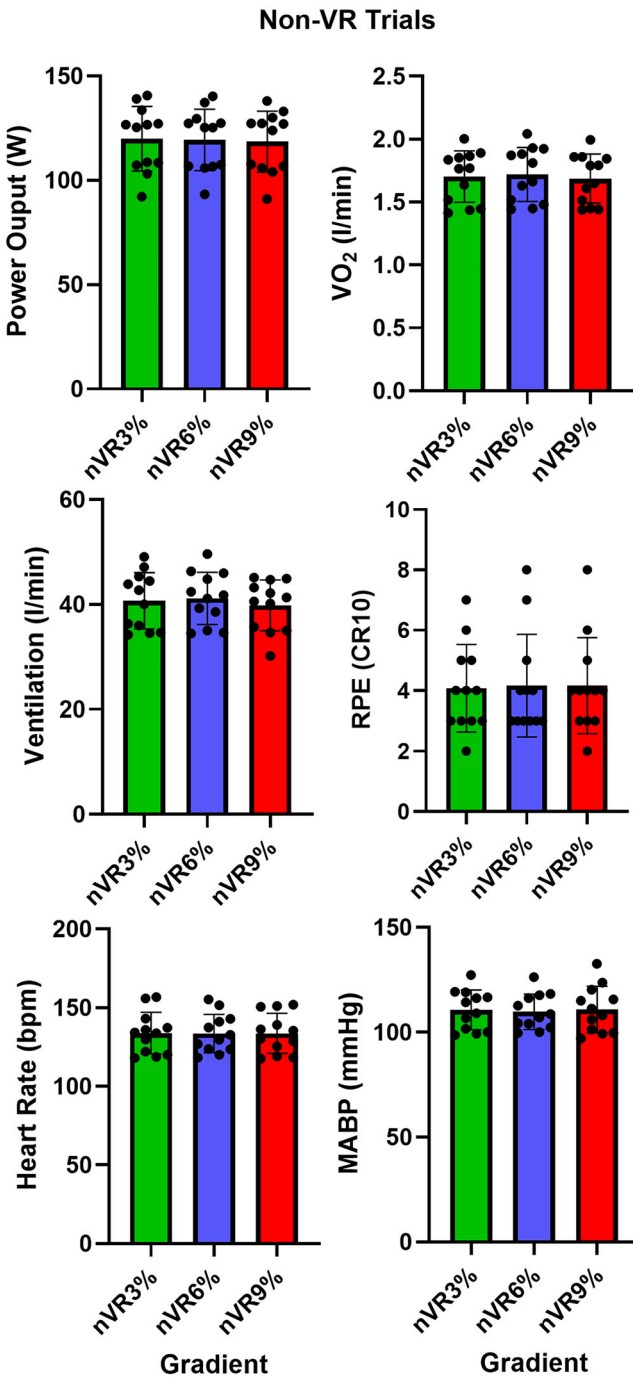

**Figure 5. The mean (±SD) power output, oxygen consumption (VO₂), ventilation, rating of perceived exertion (RPE) heart rate and mean arterial blood pressure (MABP) recorded during cycling at 3 hill gradients (nVR3%, nVR6% and nVR9%), in the laboratory 'non-VR' environment.**
Pedalling resistance was identical for each participant in each trial; the only difference was the bike tilt angle.

command in cardiovascular control (Asmussen et al., 1965; Goodwin et al., 1972; Hobbs & Gandevia, 1985; Iwamoto et al., 1987; Krogh & Lindhard, 1913; Victor et al., 1989). Many of these studies examined central command by uncoupling descending motor drive with muscular force using various techniques (e.g. neuromuscular blockade, electrically induced muscle contractions, tendon vibration). In this way these previous studies have demonstrated the existence of a feed-forward 'central command' control mechanism regulating the cardio-respiratory responses to exercise by grossly matching them to descending motor outflow. The current study adds to this work by showing that manipulation of the perception of effort using immersive VR, independent of a change in motor output, is able to regulate the cardio-vascular response to exercise. Cycling up steeper virtual hills generated significantly elevated RPE, despite no change in cycling workload. This resulted in elevated blood pressure responses and, although the HR response failed to reach statistical significance between trials (likely due to inadequate statistical power, see later), there was a significant positive correlation both between $MABP_{Diff}$ and $RPE_{Diff}$ and between $HR_{Diff}$ and $RPE_{Diff}$ (Fig. 4). This may suggest that changes in the cardio-vascular response were dependent on the changes in RPE generated by the VR. The findings that the perception of cycling workload can alter cardiovascular responses to exercise support those from previous work using hypnotic suggestion (Williamson et al., 2001). Future VR studies could measure regional cortical oxygenation during exercise (Williamson et al., 2001, 2002), but at present this remains challenging due to the presence of the VR headset. Screen-based VR environments may provide a solution, but as this is less immersive it may impact the experimental effect (Bird et al., 2024).

Despite some similarities to previous work (Williamson et al., 2001), several contrasting findings were observed. For example (i) we failed to demonstrate an effect on the ventilatory response; (ii) the experimental effect on RPE and the associated cardiovascular response decayed over time; and (iii) we failed to observe a reduction in RPE during trials, which aimed to reduce perceptions of effort. These differences could be caused by several factors in the current study design. Firstly hypnotic suggestion and immersive VR are completely different experimental models and provide different neural stimuli, so it is not unexpected for the nature, magnitude or temporality of their effects to be different. However, despite the smaller (or absent) effect sizes shown, as immersive VR aims to recreate experiences as close to real life as possible, its use likely has greater real-world application compared to hypnosis.

Secondly our current work examined differences between trials involving cycling along a virtual flat road at a low workload followed by cycling uphill with different

virtual gradients at the same elevated workload. This contrasts with previous work (Williamson et al., 2001), which assessed differences between trials involving an entirely fixed cycling workload where at some time point participants received hypnotic suggestion of cycling uphill, flat or downhill. These differences in experimental intervention could explain the differences recorded in magnitude and reliability of the experimental effect. We chose a protocol which aimed to create subtle incongruity (three different virtual uphill stages at the same elevated workload) to minimise the conscious perception of incongruence between pedalling resistance and hill gradients. Future studies should aim to optimise the incongruence and maximise the experimental effect – it is possible that under these conditions respiratory differences or less temporal effects may be observed.

Thirdly limitations in the VR environment may explain the smaller/decaying effect sizes. Using a standardised presence questionnaire (Witmer & Singer, 1998; Witmer et al., 2005) we have shown previously that the virtual environment is highly immersive (Runswick et al., 2023), but it is not a perfect representation of reality. Furthermore in the VR environment employed in the current study, when cycling uphill there was no easily visible flat horizon line in the distance on either side of the hill, which may have led to a loss of the visual perception of cycling up steeper or flatter hills. It is possible this factor may have contributed to the decay in the experimental effect, and therefore future iterations of VR cycling environments should consider providing a clear and constant visual reminder of the hill gradient.

Furthermore in contrast to our hypothesis, we did not show that cycling up less-steep virtual hills results in reduced RPE and cardiorespiratory responses during unchanged workload. A likely contributing factor may be our limited sample size failing to provide the required statistical power, and future work should aim to examine this question more closely. It is also possible that sensory feedback via skeletal muscle afferent fibres, known to be important in cardiovascular control, which remains unchanged during the trials, may have masked any experimental VR effect (Coote et al., 1971; Kaufman et al., 1983; McCloskey & Mitchell, 1972). In addition it is well established that humans tend to overestimate hill gradients (including virtual slopes), the magnitude of which increases with actual hill gradient and is associated with the physical effort thought needed to ascend (Bhalla & Proffitt, 1999; Burrow et al., 2016; Creem-Regehr et al., 2004; Proffitt et al., 2001). This might, in part, also explain why we observed differences only in RPE and the associated cardiovascular responses with the steeper slope. Further work is needed using different experimental protocols to test this hypothesis.

Despite the term 'congruent' it is important to emphasise the VR environment in **cVR** trials will never perfectly match participant expectations of a real-world environment. Additionally the pedalling resistances and virtual hill gradients (3%, 6% and 9%) in **cVR** trials were chosen as a balance between (i) maximising congruity between each resistance and its associated hill gradient, (ii) maximising the differences in sensory feedback (visual and vestibular) between the three VR hill gradients and (iii) minimising participant-conscious perception of incongruence between pedalling resistance and hill gradients in **iVR** trials. This balance of factors aimed to maximise the experimental effect. Taken together, because congruity is not perfect in **cVR** trials, visits 2–4 were included to help inform or 'condition' the participants to associate the specific hill gradients with a certain physiological challenge (i.e. 'learn' these are congruent). Future research should carefully assess the relative importance of different VR stimuli and the degree of incongruity, in combination with the minimum-required participant familiarisation to the VR environment, in the manipulation of perception in exercise exertion. Once optimised it is possible the experimental effect on RPE and cardiovascular responses will be augmented and effects on ventilatory responses might be observed. However uncovering the mechanisms responsible for exercise ventilatory control has been classically more challenging than that for cardiovascular control (Bruce, 2022; Grodins, 1981).

It is important to recognise that the differences in RPE and MABP may not have been driven by VR manipulation *per se* but due to differences in exercise bike tilt angle independent of VR immersion. Changing bike tilt angle may alter the biomechanical efficiency of cycling or require activation of other postural muscles. We consider this to be unlikely as (i) during iVR trials there were no differences in $VO_2$ and power output and, hence, no detectable change in cycling economy and (ii) there were no differences in any variable during the nVR trials where bicycle gradient was changed independently of the VR environment. Additionally in previous work in which bike tilt was maintained as constant, there was an observed change in RPE relating to congruence (Finnegan et al., 2023). This might lead us to conclude that the change in bike tilt, and its associated stimulation of the vestibular system, did not contribute to the elevations in RPE and MABP observed using the experimental VR manipulation. It is possible however that only the combination of different stimuli, such as the visual projections of hills matched to physical bike gradient, is required to create a sufficiently immersive VR environment that is able to drive the responses shown. Future research should carefully assess the relative importance of different VR stimuli in the manipulation of perception in exercise exertion.

An ability to manipulate exercise perception via VR may offer practical applications in clinical or athletic

populations. Manipulating VR environments so that perceptions of effort (or potentially pain/breathlessness) are altered may be beneficial for those with limited exercise tolerance (Buono et al., 2019; Marlow et al., 2019; O'Donnell et al., 2020). Immersive VR may also provide new opportunities for testing mechanisms theorised to underpin exercise perceptions. Experimental manipulation of higher-order cognitive factors and/or sensory information may allow feedback and feed-forward control mechanisms to be further examined (Abbiss et al., 2015; Halperin & Emanuel, 2020; Marcora, 2009). Although we did not directly test one theoretical approach, our data suggest that effort perception during exercise in part results from a complex interaction between expectations (knowledge of task difficulty, previous experiences) and sensory inputs (i.e. visual, vestibular feedback). Future research will also need to establish whether repeated exposure to incongruent stimuli will continue to manipulate exercise perceptions.

In addition other experimental approaches aiming to manipulate higher-order cognitive factors during exercise, such as shifting attentional focus, listening to music or developing mental fatigue, have been shown to impact effort perception without necessarily altering cardio-vascular responses (Brownsberger et al., 2013; Lohse & Sherwood, 2011; Potteiger et al., 2000). Differences in findings between these approaches and the current study, or work using hypnosis (Williamson et al., 2001), may reflect differences in the nature of the higher-order manipulation (e.g. utilising distractions *vs.* knowledge of task difficulty/previous experience). Indeed we have previously shown that the distraction associated with using a VR environment when compared to normal laboratory conditions was in itself insufficient to generate differences in the cardiovascular response to exercise (Runswick et al., 2023). Future work might aim to establish the mechanisms underlying the apparent differences between experimental approaches.

In conclusion the key finding from this study is that cycling up steeper virtual hills resulted in significantly elevated RPE and blood pressure despite actual workload remaining unchanged. This work therefore presents a novel model to examine the 'central command' cardio-vascular control mechanism of human exercise. These data suggest that perception of effort (RPE) and cardio-vascular responses to exercise can be manipulated experimentally via virtual hill gradient (using visual and/or vestibular cues) in a VR environment. This supports previous work showing the existence of a control mechanism which integrates perception of effort and the cardiovascular response to exercise in humans.

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

## Additional information

### Data availability statement

All original 'raw' data from which graphical and/or tabular summary data are obtained are fully available upon reasonable request.

### Competing interests

Dr Pattinson is named as co-inventor on a provisional UK patent application titled 'Use of cerebral nitric oxide donors in the assessment of the extent of brain dysfunction following injury'. Drs Pattinson and Finnegan are named as co-inventors on a provisional UK patent titled 'Discordant sensory stimulus in VR based exercise', UK Patent office application: 2204698.1, filing date: 31 March 2022. The remaining authors have no biomedical financial interests or potential conflicts of interest.

### Author contributions

All experiments were performed in R.M.B.'s laboratory at King's College London. R.M.B., G.F.R., S.L.F., M.S., K.T.S.P. and O.R.R. contributed to the conception or design of the work and the drafting the work or revising it critically for important intellectual content. R.M.B. and O.R.R. contributed to the acquisition, analysis or interpretation of data for the work.

All authors have approved the final version of the manuscript and agree to be accountable for all aspects of the work in ensuring that questions related to the accuracy or integrity of any part of the work are appropriately investigated and resolved; all persons designated as authors qualify for authorship, and all those who qualify for authorship are listed.

### Funding

The equipment used in this project was purchased through a King's Together Seed Fund awarded to Dr Bruce and Dr Runswick.

### Acknowledgements

The authors thank all the volunteers who gave up their time to participate in the study. They also thank Ms Mia Tazi for supporting pilot testing for this work. In addition they thank Dr Mike J. White for his encouragement and wise suggestions.

### Keywords

cardiovascular control, effort perception, human exercise, virtual reality

## Supporting information

Additional supporting information can be found online in the Supporting Information section at the end of the HTML view of the article. Supporting information files available:

**Peer Review History**

