## [Peer Review History · The Journal of Physiology]

Incongruent virtual reality cycling exercise demonstrates a role of perceived effort in cardiovascular control

Richard M Bruce, Gerrard F Rafferty, Sarah L Finnegan, Martin Sergeant, Kyle T S Pattinson, and Oliver R Runswick
DOI: 10.1113/JP287421

Corresponding author(s): Richard Bruce (richard.bruce@kcl.ac.uk)

The following individual(s) involved in review of this submission have agreed to reveal their identity: Hidefumi Waki (Referee #1)

Review Timeline:

Submission Date:	31-Jul-2024
Editorial Decision:	28-Aug-2024
Revision Received:	02-Nov-2024
Editorial Decision:	15-Nov-2024
Revision Received:	29-Nov-2024
Accepted:	09-Dec-2024

Senior Editor: Vaughan Macefield

Reviewing Editor: Yoshihiro Kubo

Transaction Report:

Dear Dr Bruce,

Re: JP-RP-2024-287421 "Incongruent virtual reality cycling exercise demonstrates a role of 'effort-sense' in cardiovascular control" by Richard M Bruce, Gerrard F Rafferty, Sarah L Finnegan, Martin Sergeant, Kyle T S Pattinson, and Oliver R Runswick

Thank you for submitting your manuscript to The Journal of Physiology. It has been assessed by a Reviewing Editor and by 2 expert referees and we are pleased to tell you that it is potentially acceptable for publication following satisfactory major revision.

REVISION CHECKLIST:

We look forward to receiving your revised submission.

Yours sincerely,

Vaughan Macefield
Senior Editor
The Journal of Physiology

REQUIRED ITEMS FOR REVISION

- Author photo and profile. First or joint first authors are asked to provide a short biography (no more than 100 words for one author or 150 words in total for joint first authors) and a portrait photograph. These should be uploaded and clearly labelled together in a Word document with the revised version of the manuscript. See Information for Authors for further details.

- You must start the Methods section with a paragraph headed Ethical Approval. If experiments were conducted on humans, confirmation that informed consent was obtained, preferably in writing, that the studies conformed to the standards set by the latest revision of the Declaration of Helsinki and that the procedures were approved by a properly constituted ethics committee, which should be named, must be included in the article file. If the research study was registered (clause 35 of the Declaration of Helsinki), the registration database should be indicated, otherwise the lack of registration should be noted as an exception (e.g. The study conformed to the standards set by the Declaration of Helsinki, except for registration in a database). For further information see: <https://physoc.onlinelibrary.wiley.com/hub/human-experiments>.

- Please upload separate high-quality figure files via the submission form.

- Papers must comply with the Statistics Policy: https://jp.msubmit.net/cgi-bin/main.plex?form_type=display_requirements#statistics.

In summary:

- If $n \leq 30$, all data points must be plotted in the figure in a way that reveals their range and distribution. A bar graph with data points overlaid, a box and whisker plot or a violin plot (preferably with data points included) are acceptable formats.

- If $n > 30$, then the entire raw dataset must be made available either as supporting information, or hosted on a not-for-profit repository, e.g. FigShare, with access details provided in the manuscript.

- 'n' clearly defined (e.g. x cells from y slices in z animals) in the Methods. Authors should be mindful of pseudoreplication.

- All relevant 'n' values must be clearly stated in the main text, figures and tables.

- The most appropriate summary statistic (e.g. mean or median and standard deviation) must be used. Standard Error of the Mean (SEM) alone is not permitted.

- Exact p values must be stated. Authors must not use 'greater than' or 'less than'. Exact p values must be stated to three significant figures even when 'no statistical significance' is claimed.

- Please include an Abstract Figure file, as well as the Figure Legend text within the main article file. The Abstract Figure is a piece of artwork designed to give readers an immediate understanding of the research and should summarise the main conclusions. If possible, the image should be easily 'readable' from left to right or top to bottom. It should show the physiological relevance of the manuscript so readers can assess the importance and content of its findings. Abstract Figures should not merely recapitulate other figures in the manuscript. Please try to keep the diagram as simple as possible and without superfluous information that may distract from the main conclusion(s). Abstract Figures must be provided by authors no later than the revised manuscript stage and should be uploaded as a separate file during online submission labelled as File Type 'Abstract Figure'. Please also ensure that you include the figure legend in the main article file. All Abstract Figures should be created using BioRender. Authors should use The Journal's premium BioRender account to export high-resolution images. Details on how to use and access the premium account are included as part of this email.

- Please include a full title page as part of your main article (Word) file, which should contain the following: title, authors, affiliations, corresponding author name and contact details, keywords, and running title.

EDITOR COMMENTS

Reviewing Editor: Comments to the Author (Required):

The authors the effect of the incongruence effort, given by using Virtual Reality system, on the cardiorespiratory responses. They demonstrated the existence of a control mechanism which integrates perception of exercise workload and the cardiovascular response to exercise.

I evaluate that the methodology has novelty and scientific merits to attract wide-ranged readers, even though the findings are not ground shaking.

The evaluations of the two reviewers are clearly split. The reviewer #1 evaluates the scientific merits of the work. He/she raised some major and minor points which require attention. I hopefully believe the authors can fully respond to all points and revise satisfactorily.

Reviewer #2 evaluated that the scientific merits of the new physiological findings are not beyond expectation, and not ground shaking. I would like to ask authors to intensively respond to his/her critical comments by emphasizing the scientific merits of the study and by revising the manuscript accordingly.

Senior Editor:

Thank you for submitting your manuscript to The Journal of Physiology. I have now received comments from two independent reviewers and the Reviewing Editor, all experts in the field. As you will see from their comments, there are some major issues you will need to address before we can consider the manuscript further. Please note that studies other than those mentioned have looked at the role of central command and effort in the cardiovascular and respiratory responses to actual and imagined or first-person video exercise, including increases in respiration. It would be worth citing this work. I invite you to revise the manuscript accordingly and submit point-by-point responses to the reviewers' comments. I look forward to receiving your revised manuscript in due course.

REFEREE COMMENTS

Referee #1:

The authors examined whether using a highly immersive virtual reality cycling environment to create incongruence between expected effort (virtual hill gradient) and actual effort (pedal resistance), the expectation of effort alters rating of perceived exertion (RPE) and cardiorespiratory responses to exercise independently of actual effort. They have demonstrated that cycling up a steeper virtual hill resulted in a transient elevation in RPE and blood pressure despite actual workload remaining unchanged. Their results suggest the existence of a control mechanism which integrates perception of exercise workload

('effort-sense') and the cardiovascular response to exercise and this can be experimentally manipulated by virtual reality.

The study is interesting, and the experimental procedures are well designed. It also seems to take advantage of Virtual Reality. However, some issues have been raised and need to be revised.

Major:

1. Sentences L67 to L70 (also L348 -350) cannot be understood. In particular, I do not understand the meaning of L68 (L349): "perception of exercise workload ('effort-sense')." Does the authors consider the effort sense to be the perception of the bottom-up signal, which is the internal input from mainly muscle receptors with actual movement, or the top-down signal predictably triggered by the perception of external visual information? Or, is it due to the discrepancy between the actual sense (peripheral information) and the visual sense? In the first case, the input is considered to be after the start of actual exercise, while in the second case, it should be generated when seeing a steep slope (before actually hill climbing). In any case, terms such as expectation of exercise effort, perception of exercise workload, effort-sense, and expected effort should be explained in detail. Please consider illustrating it in the figure.

2. They found that "Despite no difference in power output, there was a significantly elevated rating of perceived exertion (RPE) and mean arterial blood pressure in iVR9% compared to iVR3% and iVR6%." Because this result is most important in this study, individual data for 12 subjects should be presented (At least for blood pressure and RPE at Rest, Flat and 0.5min in Figure 3). In fact, did all 12 people have the same reaction?

3. What do the dots in Figure 4 mean? Each subject? Why mix the results of two differences (iVR9%-iVR6%, iVR9%-iVR3%)? Shouldn't the correlation between RPE and blood pressure be separated for each condition? This figure needs a detailed explanation.

4. Figure 4, L365: Why was there no difference in HR for the results in Figure 3, even though there was a significant positive correlation not only between MAP and RPE but also between HR and RPE?

Minor:

1. Blood pressure values should indicate not only mean blood pressure but also systolic and diastolic blood pressure.

2. Is it correct to understand that 1 trial at visit 5 is 1 minute rest → 1 minute flat → 3 ~ 4 minutes hill climb, VR condition (cVR, iVR, nVR) x slope (3%, 6%, 9%) = 9 trials, 15 minutes break between each trial, and therefore 180 minutes per subject (1 trial approximately 20 minutes x 9 trials)? Furthermore, if this is true, then (L243) "~during the final minute of the flat stage" is a strange expression (since there is only 1 minute to begin with)? The flow of 1 trial is difficult to understand, so I would like you to illustrate it.

3. L235: Gin et al., 2018 is not cited in the Reference list.

4. Is it that the pedal resistance is constant regardless of the tilt in the nVR condition? (Figure 5)

5. L361: CV cardiovascular?

6. Were the subjects aware that the visual stimulus and actual workload were "incongruent" in the iVR condition?

Referee #2:

Bruce and colleagues have merit for presenting a new method to manipulate effort sense using virtual reality and experimental simulations of real-life exercise. The study was carefully designed, but the effect on effort sense was small and short-lasting. Thus, the method still requires refinement to investigate physiological and pathophysiological questions regarding the neural control of cardiorespiratory responses to exercise. Most importantly, the authors stated that the study aimed to examine whether the expectation of effort alters cardiorespiratory responses to exercise independently of actual effort. This question has been investigated before by other methods, such as hypnosis, as mentioned by the authors, but also by motor imagery and observation of motor execution. Although all of these methods have limitations, as any technique does, there is already enough evidence to support that expectation of effort does alter cardiorespiratory responses to exercise independently of actual effort. Therefore, the present study does not provide a paradigm shift in the mechanistic interpretation of the neural control of cardiorespiratory responses to exercise.

END OF COMMENTS

Dear Prof Vaughan Macefield

Thank you very much for your review of the manuscript. We were pleased to read the positive and constructive comments made by the referees. We apologise for our delayed response, but please see below for our replies to the specific comments.

EDITOR COMMENTS

Senior Editor:

Thank you for submitting your manuscript to The Journal of Physiology. I have now received comments from two independent reviewers and the Reviewing Editor, all experts in the field. As you will see from their comments, there are some major issues you will need to address before we can consider the manuscript further. Please note that studies other than those mentioned have looked at the role of central command and effort in the cardiovascular and respiratory responses to actual and imagined or first-person video exercise, including increases in respiration. It would be worth citing this work. I invite you to revise the manuscript accordingly and submit point-by-point responses to the reviewers' comments. I look forward to receiving your revised manuscript in due course.

Thank you for your positive comments. We agree that we should expand our discussion of other related works as you describe, and have added these to our introduction (see paragraph 3 of introduction).

REFEREE COMMENTS

Referee #1:

The authors examined whether using a highly immersive virtual reality cycling environment to create incongruence between expected effort (virtual hill gradient) and actual effort (pedal resistance), the expectation of effort alters rating of perceived

exertion (RPE) and cardiorespiratory responses to exercise independently of actual effort. They have demonstrated that cycling up a steeper virtual hill resulted in a transient elevation in RPE and blood pressure despite actual workload remaining unchanged. Their results suggest the existence of a control mechanism which integrates perception of exercise workload ('effort-sense') and the cardiovascular response to exercise and this can be experimentally manipulated by virtual reality.

The study is interesting, and the experimental procedures are well designed. It also seems to take advantage of Virtual Reality. However, some issues have been raised and need to be revised.

Major:

1 . Sentences L67 to L70 (also L348 -350) cannot be understood. In particular, I do not understand the meaning of L68 (L349): "perception of exercise workload ('effort-sense')." Does the authors consider the effort sense to be the perception of the bottom-up signal, which is the internal input from mainly muscle receptors with actual movement, or the top-down signal predictably triggered by the perception of external visual information? Or, is it due to the discrepancy between the actual sense (peripheral information) and the visual sense? In the first case, the input is considered to be after the start of actual exercise, while in the second case, it should be generated when seeing a steep slope (before actually hill climbing). In any case, terms such as expectation of exercise effort, perception of exercise workload, effort-sense, and expected effort should be explained in detail. Please consider illustrating it in the figure.

Agreed, we have made changes to these sections to enhance clarity and consistency in terminology (see abstract and paragraph 1 of discussion). Please see clear definitions in paragraph 2 of the introduction and a discussion of the possible mechanisms which may impact perceived effort. Although we cannot be completely sure of the control mechanisms involved (and this is something hotly debated) perceived effort is likely some complex interplay between top down signalling (corollary discharge from central motor output), bottom-up signalling (sensory feedback) and also higher-order cognitive factors (such as knowledge of task difficulty and previous experience). In a similar way to the hypnosis studies, we believe we are manipulating higher order conscious perception of the task difficulty, and doing so via careful changes in sensory information (visual information from hill gradient, vestibular information from bike tilt). We hope this adds the clarity needed .

2 . They found that "Despite no difference in power output, there was a significantly

elevated rating of perceived exertion (RPE) and mean arterial blood pressure in iVR9% compared to iVR3% and iVR6%." Because this result is most important in this study, individual data for 12 subjects should be presented (At least for blood pressure and RPE at Rest, Flat and 0.5min in Figure 3). In fact, did all 12 people have the same reaction?

Agreed, we have added the individual data points to the figures. You can also derive this information from figure 4, which shown only comparison person had a lower RPE in the 9% trial and 4 people had a lower BP.

3 . What do the dots in Figure 4 mean? Each subject? Why mix the results of two differences (iVR9%-iVR6%, iVR9%-iVR 3%)? Shouldn't the correlation between RPE and blood pressure be separated for each condition? This figure needs a detailed explanation.

Figure 4 has been created by first calculating the difference (Diff) in MABP, HR and RPE recorded between trials at 0.5 min (iVR9% - iVR6% and iVR9% - iVR3%), and then the correlation between $MABP_{Diff}$ vs. RPE_{Diff} (A) and HR_{Diff} vs. RPE_{Diff} (B) was assessed. The reason why the data are combined is because we wanted to demonstrate that the significant effects in blood pressure are related to the perceptions of exercise exertion (RPE). It might be that the significant findings for MABP and RPE are simply two unrelated occurrences. This supports our argument for a mechanism involving perception of effort (manipulated by virtual hill gradient) and the CV response to exercise. We have added further details to the results section (penultimate paragraph) to explain this.

4 . Figure 4, L365: Why was there no difference in HR for the results in Figure 3, even though there was a significant positive correlation not only between MAP and RPE but also between HR and RPE?

There was s no significant difference in HR between trials because the effect size was small (generated by the participants producing lower HRs during iVR9% trials vs other trials). Despite this there was a significant correlation between changes in HR and RPE (i.e. the negative differences iVR9% and other trials may still fit a linear model well, but it will lower the mean difference between the trials, creating a non-significant mean difference). A limitation of the study was the sample size, and this low statistical power potentially contributed to the non-sig. findings in HR. We have added this point to the limitations section.

Minor:

1 . Blood pressure values should indicate not only mean blood pressure but also systolic and diastolic blood pressure.

Agreed, we have added this to the results section

2 . Is it correct to understand that 1trial at visit 5 is 1 minute rest→1 minute flat→3 ~ 4 minutes hill climb, VR condition (cVR, iVR, nVR) x slope (3%, 6%, 9%) = 9trials, 15 minutes break between each trial, and therefore 180 minutes per subject (1trial approximately 20 minutes x 9 trials)? Furthermore, if this is true, then (L243) "~during the final minute of the flat stage" is a strange expression (since there is only 1 minute to begin with)? The flow of 1trial is difficult to understand, so I would like you to illustrate it.

The flat stage was also approximately 4 minutes (same as hill). We have clarified this in the methods section.

3 . L235: Gin et al., 2018 is not cited in the Reference list.

Thank you. This has been added

4 . Is it that the pedal resistance is constant regardless of the tilt in the nVR condition? (Figure 5)

Yes, this is true. We have added text to the figure 5 legend to make this clear.

5 . L361: CV cardiovascular?

Thank you. This has been changed

6 . Were the subjects aware that the visual stimulus and actual workload were "incongruent" in the iVR condition?

The participants were not aware of the experimental manipulation. This has been made clearer in the methods.

Referee #2:

Bruce and colleagues have merit for presenting a new method to manipulate effort sense using virtual reality and experimental simulations of real-life exercise. The study was carefully designed, but the effect on effort sense was small and short-lasting. Thus, the method still requires refinement to investigate physiological and pathophysiological

questions regarding the neural control of cardiorespiratory responses to exercise. Most importantly, the authors stated that the study aimed to examine whether the expectation of effort alters cardiorespiratory responses to exercise independently of actual effort. This question has been investigated before by other methods, such as hypnosis, as mentioned by the authors, but also by motor imagery and observation of motor execution. Although all of these methods have limitations, as any technique does, there is already enough evidence to support that expectation of effort does alter cardiorespiratory responses to exercise independently of actual effort. Therefore, the present study does not provide a paradigm shift in the mechanistic interpretation of the neural control of cardiorespiratory responses to exercise.

We acknowledge the comments by reviewer #2 regarding previous studies using techniques such as hypnosis, motor imagery and observation of motor execution to examine neural control of cardiorespiratory responses to exercise.. We do not claim that our findings are ‘paradigm shifting’ but believe that using virtual reality overcomes some of the shortcomings of previous techniques, such as hypnosis, while helping clarify, confirm and expand the findings of previous studies. There are many instances in the journal where well designed research studies which have developed a novel test or experimental model have been published (which improve on previous iterations and/or provide an alternative method of assessment or examine different combinations of mechanisms), and yet produce similar data to those previously. We have tried to ensure that the language used in the manuscript is nuanced and does not overclaim on the technique, the data produced and the conclusions drawn.

The fact that our data is largely supported by previous works strengthens our arguments and helps to validate an emerging model of cardiorespiratory control in exercise. It is also important to recognise the differences between our data and others (see discussion paragraphs 3, 4, 5 and 6). We believe that the novel methodology employed in the current study and the data produced provide new insights into the underlying mechanisms involved, and provide a unique means by which to further investigate such fundamental physiological processes. Such differences highlight the novelty of our approach in comparison to those used previously.

Dear Dr Bruce,

Re: JP-RP-2024-287421R1 "Incongruent virtual reality cycling exercise demonstrates a role of perceived effort in cardiovascular control" by Richard M Bruce, Gerrard F Rafferty, Sarah L Finnegan, Martin Sergeant, Kyle T S Pattinson, and Oliver R Runswick

Thank you for submitting your revised Research Article to The Journal of Physiology. It has been assessed by the original Reviewing Editor and Referees and has been well received. Some final revisions have been requested.

REVISION CHECKLIST:

We look forward to receiving your revised submission.

Yours sincerely,

Vaughan Macefield
Senior Editor
The Journal of Physiology

REQUIRED ITEMS

- Papers must comply with the Statistics Policy: https://jp.msubmit.net/cgi-bin/main.plex?form_type=display_requirements#statistics.

In summary:

- If n {less than or equal to} 30, all data points must be plotted in the figure in a way that reveals their range and distribution. A bar graph with data points overlaid, a box and whisker plot or a violin plot (preferably with data points included) are acceptable formats.
- If $n > 30$, then the entire raw dataset must be made available either as supporting information, or hosted on a not-for-profit repository, e.g. FigShare, with access details provided in the manuscript.
- 'n' clearly defined (e.g. x cells from y slices in z animals) in the Methods. Authors should be mindful of pseudoreplication.
- All relevant 'n' values must be clearly stated in the main text, figures and tables.
- The most appropriate summary statistic (e.g. mean or median and standard deviation) must be used. Standard Error of the Mean (SEM) alone is not permitted.
- Exact p values must be stated. Authors must not use 'greater than' or 'less than'. Exact p values must be stated to three significant figures even when 'no statistical significance' is claimed.

EDITOR COMMENTS

Reviewing Editor:

Comments for Authors to ensure the paper complies with the Statistics Policy:
nothing particular

Comments to the Author:

The authors revised the manuscript satisfactorily responding to all point comments by Reviewer #1. Reviewer #1 has been satisfied with the revised version and he/she has no more specific comments.

Reviewer #2 expressed concern to the previous version about the novelty and scientific merits of the study. The authors fully responded and explained the significance and scientific merits of the study in their response letter. Reviewer #2 now evaluated the influence of the new methodology in this study, although it is incremental, in his/her opinion. I judge it has good merits worthy for reporting. Reviewer #2 raised some detailed and constructive comments to improve the manuscript further. I hopefully believe the authors can satisfactorily re-revise the manuscript in response to all points.

Senior Editor:

Thank you for submitting your revised manuscript to The Journal of Physiology. As you will see, Reviewer 1 is satisfied with your amendments but Reviewer 2 raises further concerns you will need to address before we can consider the manuscript

further. I look forward to receiving an updated version in due course.

REFeree COMMENTS

Referee #1:

The authors of this paper have provided adequate responses to my comments and concerns. I have no further questions.

Referee #2:

I thank the authors for responding to my comments and considering them when editing the manuscript. I maintain my opinion that the study is fascinating and novel methodologically but incremental physiology-wise. Therefore, I sought to provide a constructive review to improve the manuscript further, valuing the study's strengths and pondering some of its weaknesses.

Abstract's and Main text's conclusions, and Discussion's 1st paragraph

Conclusion #1 is novel and aligned with the study rationale, aims, methods, and results. However, conclusion #2 is not novel nor aligned with the main study aim. The authors themselves acknowledged in the Discussion that they did seek to test a specific mechanism. So, I recommend replacing it with the implications of such novel method, which is wealth and can make the field move forward.

Introduction

As perceived effort is a key variable in the present study, I recommend adjusting/clarifying the perceived effort physiological basis in the 2nd paragraph of the introduction, considering the following facts:

1. Multiple pieces of evidence support that the central command is the main mechanism involved in generating perceived effort, as supported by references already cited in the manuscript.
2. Recent evidence suggests reafference from muscle spindles can contribute to the perceived effort generation (for review, PMID: 37288474).
3. Muscle afferents unlikely contribute directly to the perceived effort generation (for review, PMID: 36318384). However, relatively recent evidence supports that they inhibit the central motor output, requiring greater central motor activation (e.g., PMID: 30095164). So, they likely contribute indirectly to the generation of perceived effort.

Please improve the wording in lines 93-94.

Results

The name of Figure's 3 x-axis is missing.

RPEdiff data do not seem to present a normal distribution. If so, it must be analyzed by the Spearman Rank test rather the

Pearson one. In addition, please include raw P-value in the correlation figures.

Some studies suggest that perceived exertion and respiratory frequency are associated (PMID: 26503587). So, please report respiratory frequency and tidal volume in addition to pulmonary ventilation.

Discussion

The Discussion primarily compared the present study results with those of hypnosis studies. However, many other approaches have been shown to increase or decrease perceived effort, such as attentional focus (PMID: 22102843), listening to music (PMID: 11153860), mental fatigue (PMID: 23771830), and deception (PMID: 23771830). If I am not wrong, in most or all of these, the effect on perceived effort was sustained throughout exercise, but it was not accompanied by any cardiorespiratory change. Therefore, I kindly ask you to consider other methods than the hypnosis one in the Discussion.

END OF COMMENTS

Dear Prof Vaughan Macefield

Thank you very much for the review of the revised manuscript. We were pleased to read the continued positive and constructive comments made by the referees. Please see below for our replies to the specific comments.

EDITOR COMMENTS

Senior Editor:

Thank you for submitting your revised manuscript to The Journal of Physiology. As you will see, Reviewer 1 is satisfied with your amendments but Reviewer 2 raises further concerns you will need to address before we can consider the manuscript further. I look forward to receiving an updated version in due course.

Thank you for your comments. Please see below our answers to the queries and details of amendments

REFEREE COMMENTS

Referee #1:

The authors of this paper have provided adequate responses to my comments and concerns. I have no further questions.

Thank you for your comments

Referee #2:

I thank the authors for responding to my comments and considering them when editing the manuscript. I maintain my opinion that the study is fascinating and novel methodologically but incremental physiology-wise. Therefore, I sought to provide a constructive review to improve the manuscript further, valuing the study's strengths and pondering some of its weaknesses.

Abstract's and Main text's conclusions, and Discussion's 1st paragraph

Conclusion #1 is novel and aligned with the study rationale, aims, methods, and results. However, conclusion #2 is not novel nor aligned with the main study aim. The authors themselves acknowledged in the Discussion that they did seek to test a specific mechanism. So, I recommend replacing it with the implications of such novel method, which is wealth and can make the field move forward.

Agreed. We have changed the wording of the conclusion to reflect the fact that point two is not the novel aspect and supports previous works. The new text (from abstract in this instance) is below

“These data suggest perception of effort and cardiovascular responses to exercise can be manipulated experimentally via virtual hill gradient (using visual and/or vestibular cues) in a VR environment. This work supports those previously showing the existence of a control mechanism which integrates perception of effort and the cardiovascular response to exercise in humans”.

Introduction

As perceived effort is a key variable in the present study, I recommend adjusting/clarifying the perceived effort physiological basis in the 2nd paragraph of the introduction, considering the following facts:

1. Multiple pieces of evidence support that the central command is the main mechanism involved in generating perceived effort, as supported by references already cited in the manuscript.

Agreed. We have rephrased the sentences (85-88) to highlight the stronger evidence supporting the central command evidence (see text below). However, to be clear, lines 73-81 aim to describe a traditional understanding of ‘central command’ control of the cardiorespiratory system in exercise (not of the central control of effort perception per se). We have added some text to make this clear. We would

argue that central command control of the cardiorespiratory system is typically referred to as the parallel activation of motor regions and cardiorespiratory centres but, as we argue, the involvement of effort perception should be integrated into the general understanding. However, as you rightly state above, the main aim of this paper was not to directly assess mechanisms, so we have not suggested an overarching model encompassing all variables – we have only suggested a more complex model is likely needed.

“Perceived effort during exercise (also commonly called ‘perceived exertion’ or ‘effort-sense’) is regarded as the “conscious sensation of how hard, heavy, and strenuous a physical task is”(Marcora, 2010b). The underlying mechanism is, however, still yet to be fully elucidated. debated. While the involvement of top-down signalling (corollary discharge from motor areas) is well established, the role and integration of bottom-up signalling (sensory feedback), is still debated (Abbiss et al., 2015; Amann & Secher, 2010; Marcora, 2010a, 2010b; Monjo & Allen, 2023; Pageaux, 2016).”

2. Recent evidence suggests reafference from muscle spindles can contribute to the perceived effort generation (for review, PMID: 37288474).

Agreed, this work has been added as a reference (line 89).

3. Muscle afferents unlikely contribute directly to the perceived effort generation (for review, PMID: 36318384). However, relatively recent evidence supports that they inhibit the central motor output, requiring greater central motor activation (e.g., PMID: 30095164). So, they likely contribute indirectly to the generation of perceived effort.

Agreed, we have rephrased the sentences (85-88) as mentioned above. Although, again to be clear, when we specifically mention muscle afferent feedback as a sensory signal, it is in relation to cardiorespiratory control where it is well understood to play an important role (see citations line 74).

Please improve the wording in lines 93-94.

Agreed, see changes (and below)

“Indeed, innovative experimental designs have shown that imagined exercise (Williamson et al., 2002), or watching video footage of first-person “point-of-view” exercise (Brown et al., 2013), can generate cardiorespiratory responses approximating those produced during active exercise”

Results

The name of Figure's 3 x-axis is missing.

Thank you, this has been added

RPEdiff data do not seem to present a normal distribution. If so, it must be analyzed by the Spearman Rank test rather the Pearson one. In addition, please include raw P-value in the correlation figures.

Agreed. This has been amended

Some studies suggest that perceived exertion and respiratory frequency are associated (PMID: 26503587). So, please report respiratory frequency and tidal volume in addition to pulmonary ventilation.

Agreed, this has now been added to the result section. No significant interaction effect was shown, but as you suggest this is worth stating

Discussion

The Discussion primarily compared the present study results with those of hypnosis studies. However, many other approaches have been shown to increase or decrease perceived effort, such as attentional focus (PMID: 22102843), listening to music (PMID: 11153860), mental fatigue (PMID: 23771830), and deception (PMID: 23771830). If I am not wrong, in most or all of these, the effect on perceived effort was sustained throughout exercise, but it was not accompanied by any cardiorespiratory change. Therefore, I kindly ask you to consider other methods than the hypnosis one in the Discussion.

Agreed. A final paragraph has been added discussing this interesting point of difference. See text below

“In addition, other experimental approaches aiming to manipulate higher-order cognitive factors during exercise, such as shifting attentional focus, listening to music, or developing mental fatigue, have been shown to impact effort perception without necessarily altering cardiovascular responses (Brownsberger et al., 2013; Lohse & Sherwood, 2011; Potteiger et al., 2000). Differences in findings between these approaches and the current study, or work using

hypnosis(Williamson et al., 2001), may reflect differences in the nature of the higher-order manipulation (e.g. utilising distractions vs. knowledge of task difficulty/previous experience). Indeed, we have previously shown that the distraction associated with using a VR environment when compared to normal laboratory conditions was in itself insufficient to generate differences in the cardiovascular response to exercise (Runswick et al., 2023). Future work might aim to establish the mechanisms underlying the apparent differences between experimental approaches.”

Dear Dr Bruce,

Re: JP-RP-2024-287421R2 "Incongruent virtual reality cycling exercise demonstrates a role of perceived effort in cardiovascular control" by Richard M Bruce, Gerrard F Rafferty, Sarah L Finnegan, Martin Sergeant, Kyle T S Pattinson, and Oliver R Runswick

We are pleased to tell you that your paper has been accepted for publication in The Journal of Physiology.

Yours sincerely,

Vaughan Macefield
Senior Editor
The Journal of Physiology

If you would like to receive our 'Research Roundup', a monthly newsletter highlighting the cutting-edge research published in The Physiological Society's family of journals (The Journal of Physiology, Experimental Physiology, Physiological Reports, The Journal of Nutritional Physiology and The Journal of Precision Medicine: Health and Disease), please click this link, fill in your name and email address and select 'Research Roundup':
<https://www.physoc.org/journals-and-media/membernews>

- You can help your research get the attention it deserves! Check out Wiley's free Promotion Guide for best-practice recommendations for promoting your work at: www.wileyauthors.com/eeo/guide. You can learn more about Wiley Editing Services which offers professional video, design, and writing services to create shareable video abstracts, infographics, conference posters, lay summaries, and research news stories for your research at: www.wileyauthors.com/eeo/promotion.

EDITOR COMMENTS

Reviewing Editor:

Referee #2 has been satisfied with the revisions, and has no further comments.

Senior Editor:

Thank you for attending to these additional comments. The reviewer and reviewing editor are satisfied with your amendments so I'm pleased to report that your manuscript is now considered acceptable for publication.

REFEREE COMMENTS

Referee #2:

I have no further comments.